# The Pigments of the Painter Fleury Richard (1777–1852), a Model for Multidisciplinary Study

Davy Carole [1],*, Erika Wicky [2],*, Amina Bensalah-Ledoux [3], Stéphane Paccoud [4], Cécile Le Luyer [3], Anne Pillonnet [3] and Gérard Panczer [3]

1    Laboratoire des Multimatériaux et Interfaces, UMR 5615 Université Lyon1/CNRS, Université de Lyon, Campus de la Doua, 69622 Villeurbanne, France
2    Laboratoire de Recherche Historique Rhône-Alpes, UMR 5190 Université Lyon 2/CNRS, CEDEX 07, 69363 Lyon, France
3    Institut Lumière Matière, UMR 5306 Université Lyon1/CNRS, Université de Lyon, Campus de la Doua, 69622 Villeurbanne, France; amina.bensalah-ledoux@univ-lyon1.fr (A.B.-L.); cecile.urlacher-le-luyer@univ-lyon1.fr (C.L.L.); anne.pillonnet@univ-lyon1.fr (A.P.); gerard.panczer@univ-lyon1.fr (G.P.)
4    Musée des Beaux-Arts de Lyon, 69001 Lyon, France; stephane.paccoud@mairie-lyon.fr
*    Correspondence: davy.carole@univ-lyon1.fr (D.C.); erika.wicky@gmail.com (E.W.)

**Abstract:** Fleury Richard was a colorist painter of the early 19th century. He practiced the oil technique inspired by the Renaissance at a time when advances in chemistry were introducing many new synthetic pigments. His color-mixing cabinet has been kept intact at the Musée des Beaux Arts de Lyon. This original study is based on the analysis of more than 40 color powders using different spectroscopic techniques (X-ray diffraction (XRD), Fourier transform infrared spectroscopy (FTIR), and Raman spectroscopy), color index estimation, and the comparison of the results obtained from three pictural works painted by the artist. It allows us (i) to identify and reference the pigmented powders and pictural choices in connection with historical manuscripts describing the artist's practice, and (ii) to identify the most judicious analysis methods and question the difficulty of analyzing paintings in a non-destructive way, where pigments are put into a matrix and mixed.

**Keywords:** pigments; oil painting technique; composition; paintings; color box

## 1. Introduction

Making the title of "colorist" given to him by his master Jacques Louis David his own (and later by the art critics of his time), the Lyon-based painter Fleury Richard (1777–1852) placed color at the heart of his artistic practice. The great success of his 1802 painting *Valentine of Milan mourning the death of her husband the Duke Louis of Orléans* is based both on the novelty of the "troubadour" style, which borrows its subjects from the medieval period [1], and on the highly sophisticated treatment of color. Indeed, a green taffeta curtain, drawn in front of a window decorated with stained glass in the background, gives the illusion of being pierced by light. Fleury Richard experimented and worked on the effects of transparency, inspired by the technique of the Dutch masters of the Golden Age, seeking not only colorful effects but also the persistence of colors in the long term, a major concern in this period when experimentation was multiplying [2]. In his memories, which remain in manuscript form, the painter confides not only his attachment to color as an artist, but also the technical difficulties posed by his requirements as a colorist, his recourse to little-known materials that were difficult to work with, as well as his hope that his work as a colorist preserves his name and his memory. In this way, he is part of his time, for, as Charlotte Guichard notes [3], "the notion of color in the late 18th century was constructed [ . . . ] at the boundaries of the physics of light, the chemistry of pigments, and the theory of art" (from French: « La notion de couleur à la fin du 18e siècle se construit [ . . . ] aux confins de la physique de la lumière, de la chimie des pigments et de la théorie de l'art »).

Fleury Richard's passion for color appears in several works, some of which are exhibited in the collections of the Musée des Beaux-Arts de Lyon, such as *Vert-Vert* (1804), *The Painter and his family* (1817 or 1822) or *Tasso in his prison visited by Montaigne* (1821). His work on color is singular, first of all, because it mixes old techniques (the painter evokes in his memories a "system of colors and execution that I borrowed from the Dutch [4]") with new materials born of scientific research, such as Scheele's green, which appeared in 1778, or Cobalt blue in 1805. Furthermore, Fleury Richard was at the crossroads of two eras: one in which painters still mixed their pigments themselves with an additive, a binder, or a siccative (generally linseed oil and turpentine), or even ground them themselves to obtain the desired textures, and one in which the production of colors was beginning to be industrialized and color merchants were offering artists more and more elaborate materials, reducing the time spent by the artist in the preparation of colors. As Pernety [5] wrote in his *Dictionnaire portatif de peinture sculpture et gravure* (Portable Dictionary of Painting): "All natural or false colors are found at the Epiciers, Marchands de couleurs. They sell them wholesale or retail, either in stone, loaves or powder, or crushed in oil" (from French: « *Toutes les couleurs naturelles ou factices se trouvent chez les Épiciers, Marchands de couleurs. Ils les vendent en gros ou en détail, soit en pierre, en pains ou en poudre, soit broyées à l'huile* »). Indeed, the color merchants not only sold artists the materials of their art, but also prepared them thanks to the work of the color grinders that reduced the pigments to powder, sometimes adding oil or materials likely to improve them, such as alum to madder [6]. After the abolition of the guilds in 1791, the number of color merchants continued to grow, as did the range of products marketed, inaugurating a new relationship between painters and their materials [7].

The study of Fleury Richard's pictorial technique is of double interest as it provides information on his singular practice as a colorist and a transitional period in the history of the material culture of painting. In addition to Fleury Richard's works and memorabilia, the Musée des Beaux-Arts de Lyon owns a precious piece of furniture that can shed light on Fleury Richard's work as a colorist: his color-mixing cabinet, an important piece of the Fleury Richard's collection that the museum has held since 1988. An essential element of the art of painting, the painter's cabinet, like its transportable counterpart, the color box, contains all the material necessary for this practice. In this case, it is a piece of furniture made of wood, with eight drawers holding containers in which we find material for grinding colors and pestles also intended for this purpose, brushes and palettes, tin tubes containing gouache, glass bottles containing Garance lake, as well as more than 110 parcel papers containing pigmented powders [8] (Figure 1). Usually consisting of a sheet of newspaper or a folded letter, these parcel papers are usually annotated with the painter's hand. The inscriptions designate the colored powders they contain, sometimes using common formulas such as *brun de momie* (mummy brown) or *blanc de plomb* (lead white); sometimes using references to personal judgments such as *laque magnifique* (magnificent lake), or *laque superbe* (superb lake) or to characteristics such as *laque bleue de garance inaltérable* (unalterable madder blue lake); or even using references to contemporaries such as the *couleur d'Appiani* (Appiani's color) named after the Italian painter (1754–1817), the *grand noir de M. Dechazelle* (Great black of M. Dechazelle) in reference to Pierre Toussaint Dechazelle (1752–1833), a silk manufacturer from Lyon and notable art lover who was a great supporter of Fleury Richard and who could have provided him these pigmented powders. Finally, several parcel papers specify the origin of the pigmented powders by mentioning the names of color merchants such as Antheaume or Belot (spelled Bellot), merchants in Paris [9]. In addition, the fold of the pigmented powder *Laque de Gaude* (Gaude lake) has the delivery note "*A Monsieur Richard au Palais St Pierre à Lyon*" ("To Monsieur Richard at the Palais St Pierre in Lyon"). It was in 1809 that Fleury Richard set up his workshop at the Palais Saint-Pierre, a workshop that was offered to him by the city of Lyon as an acknowledgment to the notoriety he brought to the city. The pigments are therefore obviously those that the artist used for his paintings between 1810 and 1820, as confirmed by the date (1816) of a page of the *Journal des débats politiques et littéraires* used as a pouch.

These indications are therefore consistent with the years 1810 to 1820, corresponding to the use of raw pigments in the color furniture. Fleury Richard painted until 1824, when he found it increasingly difficult due to a nervous illness. Since his health did not allow him to continue his career beyond the 1820s, we can deduce that the lead and tin paint tubes [10], which in the 1830s replaced the pig bladders previously used to transport the pigmented powders mixed with oil [11], only later joined the pigmented powders in the color cabinet and were not part of his palette.

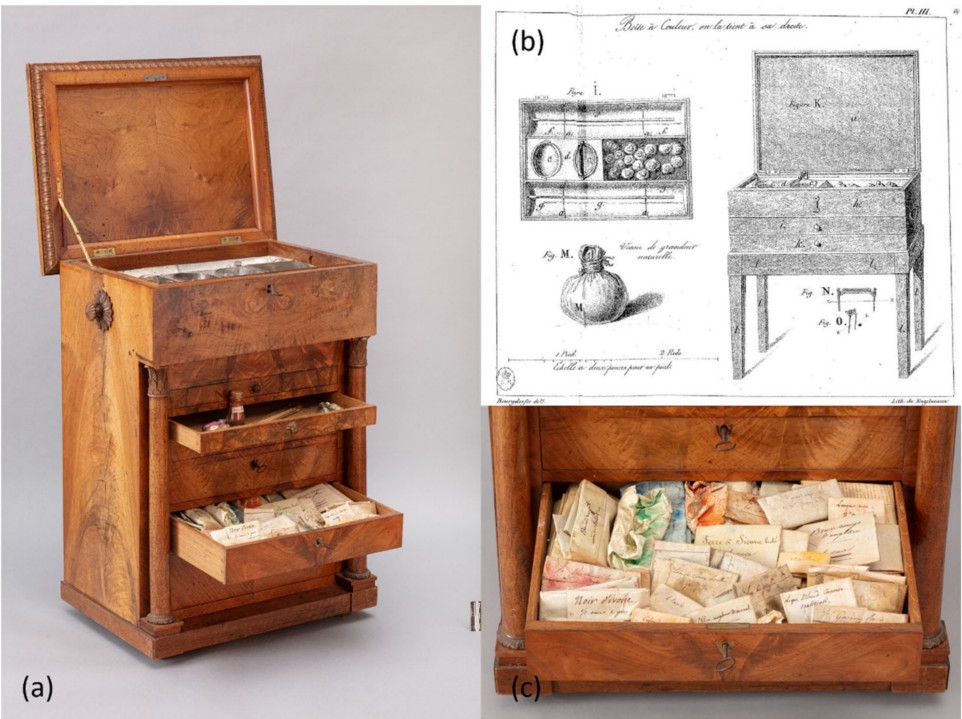

**Figure 1.** Fleury Richard's color-mixing cabinet (1810–1820): (**a**) open; (**b**) an engraving of a color box (Plate III, Bouvier, 1827; Gallica/BNF) [11] for comparison; (**c**) a drawer with part of the pigmented powders (© Musée des Beaux-Arts de Lyon, 1988-4-I-2).

Furthermore, as the names of these pigmented powders are not sufficient to identify their nature with confidence, analyses are necessary. Thus, the purpose of this study is (1) to determine the composition of the preserved pigmented powders (raw pigments or mixtures, mineral or organic pigments, natural or synthetic pigments) on the site or from samples, using different analytical techniques, as well as (2) to identify the pigments used in some paintings belonging to the collection of the Musée des Beaux-Arts de Lyon among those present in Fleury Richard's cabinet. To do so, we used different complementary spectroscopies on the raw pigmented powders from which we had the chance to take sufficient quantities (FTIR, DRX, optical absorption, XRF and Raman spectroscopies). We focused on the analysis of mineral pigments, but since organic ones are an integral part of Fleury Richard's palette, they are also presented in this work. In addition, concerning the paintings, from which we could not take samples, only non-invasive portable XRF analysis was used, as well as color analysis. Similar studies have been conducted on pigments from ancient works of art using a combination of similar spectroscopic techniques [12–14].

We have a rare opportunity to characterize the original pigments dating from the 18th century, preserved as they were for more than 200 years in Fleury Richard's cabinet, and to compare them with his paintings.

## 2. Materials and Methods

### 2.1. Materials

2.1.1. Raw Pigmented Powders

A selection of 43 representative pigmented powders out of the 110 stored in Fleury Richard's color-mixing cabinet was analyzed.

2.1.2. Paintings

Three of Fleury Richard's paintings were selected and analyzed on the site of the Musée des Beaux-Arts de Lyon by non-destructive techniques: X-ray fluorescence (XRF) and colorimetry. They are:

- *Princess Elizabeth, sister of King Louis XVI, distributing milk* (c. 1816), with analysis of red, green, white, and blue zones;
- *Henri IV at Gabrielle d'Estrées* (c. 1810–1812), with analysis of red, green, and white zones;
- *The Painter and his family* (c. 1817 or 1822), with analysis of red, green, and white zones.

### 2.2. Analytical Techniques

All 43 selected pigmented powders were systematically analyzed by X-ray fluorescence (XRF) on-site, and by X-ray powder diffraction (XRD) in the laboratory, to determine their compositions and structures (SI1). XRD is an efficient method to characterize the structure of crystallized materials and identify them. Since amorphous materials (not or slightly crystallized) are more difficult to determine by XRD, XRF allows for identifying their chemical composition by detecting the intermediate to heavy elements beyond magnesium. The lighter elements (carbon, hydrogen, nitrogen, oxygen), which compose for the most part the organic vegetal or animal compounds, cannot be detected by this method. When these two complementary and combined methods did not allow for a conclusion, we used Fourier transform infrared spectroscopy (FTIR) and/or Raman spectroscopy to try to identify the organic or poorly crystallized compounds in particular. The results of the analysis of the different data obtained thanks to these complementary spectroscopic methods are summarized in the following Tables and provided in the Supplementary Materials (SI1 and SI2). In parallel, the colorimetric analysis of the pigmented powders was carried out, and the results were confronted with the measurements of the colored areas on the artist's canvas (SI2).

2.2.1. X-ray Fluorescence (XRF)

The elemental chemical analysis of the pigmented powders is obtained by measuring the elements with atomic number $Z \geq 11$ (magnesium). The measurements were conducted directly on the parcel papers containing the pigmented powders, since the wrapping paper (cellulose) composed of light elements does not hinder the measurement. However, the porosity of the powders allowed only a qualitative analysis. This non-destructive elemental analysis method is based on the energy of X-rays emitted during the interaction of incident X-rays with the core electrons of the material atoms. The instrument used is a Thermo Niton XL3t 980 GOLDD+ XRF analyzer (Waltham, MA, USA) (maximum X-ray energy 50 kV, Ag anode). The lateral spatial resolution is 3 mm (diameter), the X-ray detection depth depends on the chemical composition (from 0.1 to 10 mm), and the analysis time is 120 s. The "Minerals" mode was use after a preliminary calibration on NIST610 and 612 standards of known compositions. It consists in 4 consecutive acquisition conditions allowing for detecting the full range of elements: 50 kV with Al filter (for intermediate elements), 50 kV with Mo filter (for high elements), 20 kV with Cu filter (for low elements), and 8 kV without filter (for light elements). The accuracy is variable depending on the elements and the matrix analyzed. In the case of porous powder pigments as well as of paintings (multi-layer structure), only qualitative analyses could be obtained.

2.2.2. X-ray Powder Diffraction (XRD)

Based on Bragg's law, X-ray powder diffraction is a non-destructive analysis allowing to identify the inter lattice distances and thus the nature of the crystallized materials. It is

carried out on compacted powder of few mg of material at the *Centre de diffractométrie Henri Longchambon* (Lyon) using a Bruker D8 Advance diffractometer (Billerica, MA, USA) (Bragg-Brentano geometry). The analysis parameters are as follows: acquisitions performed with a current of 40 mA and a voltage of 40 kV using the Kα1 line of copper with wavelength 1.5418 Å. The angular range is between 5 and 70° (2θ), with an acquisition step of 0.0205° and a quite rapid acquisition time of 192 s (e.g., Figure 2a,b).

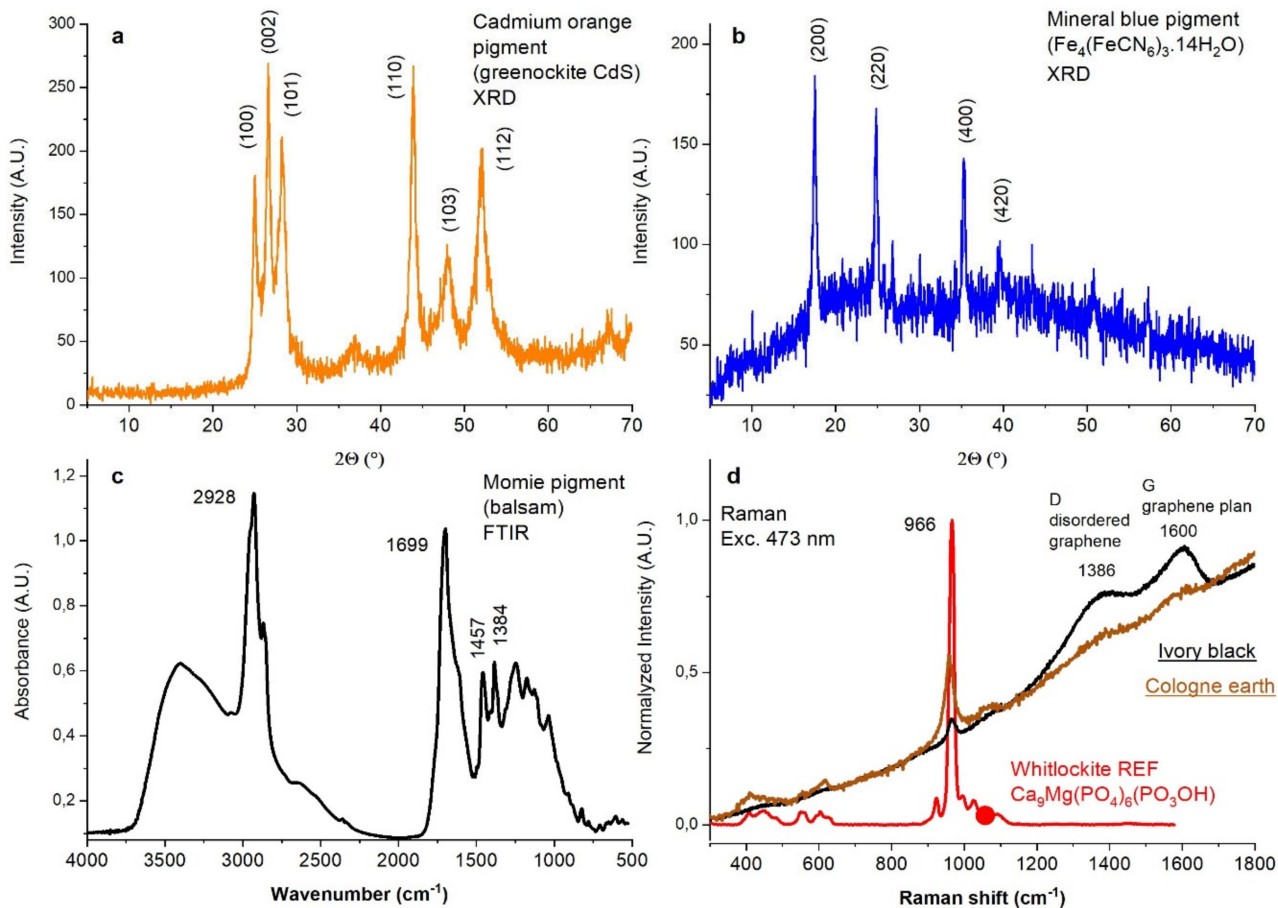

**Figure 2.** Some representative spectra (compared with references) of (**a**) the Cadmium pigmented powder (XRD); (**b**) the Mineral blue powder (FTIR); (**c**) the Momie powder (XRD); and (**d**) the Ivory black and Cologne earth powders (Raman).

### 2.2.3. Infrared Absorption (FTIR)

A sample of 1 mg of each pigment was grounded in an agate mortar and mixed with 200 to 300 mg of potassium bromide (KBr) powder. The mixture was pelleted under hydraulic press. The Fourier transform infrared spectrometer used is the FTIR Thermo Nicolet IS20 MID-IR DTGS of CECOMO, Center of vibrational spectrometry of the Institut Lumière-Matière (UMR5306, Université Lyon 1). FTIR spectra were obtained in transmission mode using the KBr pellet technique and driven by the OMNIC software. The spectral resolution is 0.4 cm$^{-1}$, and the number of scans is 30 for the background and the sample measurements. The spectra are compared to a reference database (e.g., Figure 2c). The infrared spectroscopy is not disturbed by a possible fluorescence phenomenon as it appears in Raman spectroscopy; it has a low spectral resolution and a good efficiency for organic compounds.

### 2.2.4. Micro-Raman

Micro-Raman spectrometry is a non-destructive optical analytical method based on the inelastic scattering of light on crystalline or non-crystalline materials, as well as organic

or mineral materials. It allows for evidencing atomic or molecular group vibrations of the material structure. The structural signature allows for identifying the nature of the analyzed material (e.g., Figure 2d). The use of different laser sources enables the selection of the excitation wavelength that generates little or no fluorescence (473, 532, and 633 nm). The laser power must however be modulated in order not to induce alterations due to the laser beam. The following microspectrometers were used: a Horiba Jobin-Yvon, LabRAM Aramis (473 and 633 nm), and a LabRAM HR Evolution (532 nm) (Longjumeau, France). The measurements were carried out with power laser at the surface of 17 mW on some mg of powder. Objective magnification ×50 with numerical aperture of 0.45, lateral resolution of 1.3 to 1.7 μm, and axial resolution 9 to 12.0 μm (depending on the wavelength) were used.

2.2.5. Color Images Analyses

The purpose of this section is to compare the color of the folded pigmented powders with the color of the paintings. The colorimeter is usually used to determine the colorimetric coordinates of a color. It is a non-destructive analysis that allows for associating L*a*b* coordinates with the color of the materials. It requires placing the colorimeter device in a flat position in contact with the materials. To perform the measurement of each pigmented powder, a layer of the raw pigment was deposited on a sheet of white paper containing no bleaching agents. Thus, the white paper was used as a reference. The colorimeter used was a portable digital PCE-CSM 1 (Southampton, UK) with an aperture of 6 mm, a D65 light source, and an observation angle of 10°. The measurements of the colorimetric indices L*, a*, and b* were then recorded over an average of 5 measurements. The covering power of each pigment influenced the measurements in their level of clarity. The obtained results are presented in SI2.

However, it was not possible to proceed this way on Fleury Richard's paintings as it might have damaged the pictorial layer. Indeed, to measure the colors correctly on the paintings, it would have been necessary to place the colorimeter flat in contact with the canvas, but the surface of the painting is rough, and obviously, no pressure is conceivable due to the risk of damaging the pictorial layer. To counteract this problem and obtain information on the colors of the paintings, we conducted a digital measurement of the colors from high-definition photos of the paintings. Thanks to GIMP 2.10 software, we were able to determine the RGB coordinates of the same areas analyzed by X-ray fluorescence. The corresponding coordinates L*, a*, and b* in the CIELAB color space were then obtained by NiX Color Sensor software (see SI2). For comparison, we proceeded in the exact same way with the photos of the pigmented powders. The photos were taken in the same conditions with a Canon EOS 750D camera (Canon EFS 15–55 mm objective lens) (Tokyo, Japan) under diffuse white LED light and with a color reference card for calibration.

Considering the correction with respect to the color charts used when taking the photos, we can reasonably compare the hues (a* and b*) of the pigmented powders and the measured areas on the three tables. The white and black pigmented powders were not measured.

**3. Results**

The results are organized by color, presenting first the analysis and identification of the raw pigmented powders, then the analysis of the pictural layers by comparing the analyzed areas to the raw pigmented powders.

*3.1. Raw Pigmented Powders*

The identification of the compounds of the raw pigmented powders could be made thanks to the coupling of the various analytical techniques. The tables below present the name of each pigment, the possible presence of additives in the raw pigmented powder (SI1), and finally its colorimetric coordinates (SI2). It should be noted that some parcel papers had no mention of the pigment name (*no name*).

### 3.1.1. Yellow and Orange Powders

Most yellow and orange crude pigmented powders are mixtures of pigments and additives (Table 1). In these powders, we identified crocoite ($PbCrO_4$), goethite $FeO(OH)$, jarosite ($KFe_3(SO_4)_2(OH)_2$), asisite and leadhillite, orpiment ($As_2S_3$), greenockite (CdS) (Figure 2a), and minium ($Pb_3O_4$) (SI1). All of these pigments are minerals and present in nature. Some of them present additive phases to increase the mass of the coloring matter without reducing the coloring power. The manufacturing cost of this mixture is therefore lower than that of a pure pigment. It is worth noting that in the pigments used by Fleury Richard, there is no tin lead yellow ($Pb_2SnO_4$) or Naples yellow ($Pb_2Sb_2O_7$), although these were widely used in painting in his time [15]. With regard to the pigmented powders originally named *Lacque de Gaude* (Gaude Lake) and *Lacque jaune* (yellow lake), this mentioned lake often refers to organic pigments of vegetable or animal origin. Gaude is a plant better known as *reseda luteola* (or dyer's rocket), which allowed for obtaining a yellow pigment. Although we have not been able to identify this compound by the analytical techniques used, it is very likely that these two powders with the name of the lake correspond to this substance. The powder named *Jaune clair* (light yellow) remains unidentified by the various XRD, Raman, and FTIR methods. The compound can be an organic dye; note, however, that the X-ray fluorescence revealed the presence of iron.

**Table 1.** Raw yellow and orange powders (pigments and additives) and their identification (in *italic* font: names written on the parcel papers; in normal font: names given by the authors in absence of mentions; in **bold** font: the elements detected by XRF; see SI1).

| Name | Pigment | Additive | L*; a*; b* |
|---|---|---|---|
| Petite pierre jaune [Small yellow stone] | Asisite $\textbf{Pb}_7O_8Cl_2$, leadhilite $\textbf{Pb}_4(\textbf{SO}_4)(CO_3)_2(OH)_2$ | Hydrocerussite $2\textbf{Pb}CO_3 \ \textbf{Pb}(OH)_2$ | 89 ; −1 ; 46 |
| *Jaune de Vienne* [*Vienna yellow*] | Crocoite $\textbf{PbCr}O_4$ | Cerussite $\textbf{Pb}CO_3$ | 87 ; 1 ; 73 |
| *Crôme Citron* [*Lemon chrome*] | Crocoite $\textbf{PbCr}O_4$ | | 88 ; 1 ; 70 |
| *Pierre Jaune* [*Yellow Stone*] | Goethite $\textbf{Fe}O(OH)$ | Quartz $\textbf{Si}O_2$, kaolinite $Al_2Si_2O_5(OH)_4$ | 72 ; 19 ; 53 |
| *Ocre jaune* [*Yellow Ochre*] | Goethite $\textbf{Fe}O(OH)$ | quartz, $\textbf{Si}O_2$ | 52 ; 22 ; 52 |
| *Ocre de Rue (or Ru)* [*Ru ochre*] | Goethite $\textbf{Fe}O(OH)$ | Quartz $\textbf{Si}O_2$, barite $\textbf{Ba}SO_4$ | 72 ; 26 ; 56 |
| *Jaune Marron* [*Brown Yellow*] | Jarosite $\textbf{KFe}_3(SO_4)_2(OH)_2$ | | 55 ; 42 ; 54 |
| *Orpin jaune* [*orpiment*] | Orpiment $\textbf{As}_2\textbf{S}_3$ | | 91 ; −1 ; 62 |
| Sans nom 'marron' [No name brown] | Arsenolite $\textbf{As}_2O_3$, orpiment $\textbf{As}_2\textbf{S}_3$ | | 55 ; 35 ; 51 |
| *Crôme orange* [*chrome orange*] | Crocoite $\textbf{PbCr}O_4$ | | 86 ; 5 ; 71 |
| *Cadmium* | Greenockite $\textbf{CdS}$ | | 67 ; 29 ; 61 |

**Table 1.** *Cont.*

| Name | Pigment | Additive | L*; a*; b* |
|---|---|---|---|
| Sans nom 'orange' [No name Orange] | Minium $Pb^{2+}_2Pb^{4+}O_4$ or $Pb_3O_4$ | | 62 ; 48 ; 48 |
| *Lacque de Gaude* [Gaude lake] | *Reseda Luteola* Luteolin? | Quartz $SiO_2$ | 82 ; 10 ; 61 |
| *Laque Jaune* [Yellow lake] | *Reseda Luteola* Luteolin? | Quartz $SiO_2$, gypsum $CaSO_4.2H_2O$ | 72 ; 14 ; 53 |
| *Jaune clair n°1* [Light yellow] | Organic lake? | | 88 ; 4 ; 52 |

### 3.1.2. Red Powders

The red powders (Table 2) revealed the presence of hematite and vermilion (or cinnabar) as well as cochineal red, an organic pigment of animal origin (SI1). However, there are some difficulties concerning the analysis of two powders, *laque de Garance* (madder lake) and *galet rouge* (red pebble). The madder lake contains an organic pigment of vegetable origin. Indeed, this pigment is extracted from the roots of madder, a plant that was widely cultivated in the 18th century in the south of France [16]. In the pigment extracted from the roots of the plant, there are more than 30 coloring molecules, in variable contents and whose large number makes the analysis very complicated. It can be assumed that *laque de Garance* and *galet rouge* pigments correspond to pigments of this type. It should be noted that alizarin is the name of one of the most abundant coloring molecules in madder lake. The red powders analyzed do not show any additive except for the red pebble, which probably contains gibbsite and chalk (calcite). As is known, X-ray diffraction appeared to be the most suitable technique for detecting inorganic compounds.

**Table 2.** Raw red and orange powders (pigments and additives) and their identification (in *italic* font: names written on the parcel papers; in normal font: names given by the authors in the absence of mentions; in **bold** font: the elements detected by XRF; see SI1).

| Name | Pigment | Additive | L*; a*; b* |
|---|---|---|---|
| *Brun Rouge d'angleterre d'Antheaume* [Antheaume's English redbrown] | Hematite $Fe_2O_3$ | | 33 ; 50 ; 41 |
| *Rouge Mars* [Mars red] | Hematite $Fe_2O_3$ | | 29 ; 45 ; 41 |
| *Vermillon* [Vermillion] | Vermillon **HgS** | | 47 ; 61 ; 43 |
| *Vermillon de la Chine* [Vermillion from China] | Vermillon **HgS** | | 39 ; 57 ; 47 |
| *Laque de Garance pour M^er* [Madder lake] | Alizarin? | | 27 ; 42 ; 32 |
| Galet rouge [Red pebble] | Alizarin? | Gibbsite **Al**$(OH)_3$, calcite **Ca**$CO_3$ (chalk) | 47 ; 50 ; 39 |
| *Laque magnifique* [Magnificent lake] | Carminic acid (cochineal) | | 41 ; 60 ; 43 |

### 3.1.3. Blue Powders

Most blue powders (Table 3) have been identified as either Prussian blue $Fe_4[FeCN_6]_3.14H_2O$ (Figure 2b), which was discovered by accident by the color merchant Johann Jacob Diesbach in Berlin in 1706 [17], or cobalt blue $Al_2CoO_4$, for which the manufacturing process was developed by the French chemist Thénard in 1802 [18–20]. Iron and cobalt are the coloring elements of the two pigments, respectively. Some of the blue pigmented powders are mixtures and include gypsum or alunite (*sans nom bleu*/No name blue), silica (*Bleu de Vienne*/Vienna blue), and lead or calcium carbonates (*Cobalt*, handwritten term on the parcel paper).

Since madder lake is a red pigment, the name *Laque bleue de Garance* (blue madder lake) is rather surprising for a pigment that turns out to be Prussian blue. However, the term madder blue is used in several references from the beginning of the 20th century as a synonym for artificial ultramarine [21,22] or Bleu Guimet (Guimet's blue), discovered in 1828 and therefore not used by Fleury Richard because it was used after he stopped painting. It seems therefore that the name *Laque bleue de Garance* was used before 1928 to designate Prussian blue. Furthermore, we can note the absence of Bleu de céruléum (Cerulean blue) ($Co_2SnO_4$) in Fleury Richard's palette.

In the case of the pigmented powders *Bleu de Prusse* (Prussian blue) and *Bleu mineral* (Mineral blue), the Prussian blue compound was detected by XRD (Figure 2b), FTIR, and Raman (SI1). Prussian blue is usually obtained with very small grains [23], which can provide a low-intensity diffractogram compared to well-crystallized phases such as gypsum or alum (SI1). The most difficult pigment to identify was the powder with the inscription *Cobalt*. X-ray fluorescence detected the element Co, but no cobalt-based phase was detected by XRD, Raman, or FTIR. Hydrocerussite $Pb_3(CO_3)_2(OH)_2$ or leadhillite $Pb_4(CO_3)_2(SO_4)(OH)_2$ are possible identified phases by FTIR, and lead was detected by XRF. Its colorimetric index is in good agreement with the *Bleu de Vienne* (Vienna blue) pigment, which is a cobalt oxide.

**Table 3.** Raw blue powders (pigments and additives) and their identification (in *italic* font: names written on the parcel papers; in normal font: names given by the authors in the absence of mentions; in **bold** font: the elements detected by XRF; see SI1).

| Name | Pigment | Additive | L*; a*; b* |
|---|---|---|---|
| *Bleu de Vienne* [Vienna blue] | Cobalt blue $\mathbf{Al_2CoO_4}$ | Silica $\mathbf{SiO_2}$ | 40 ; 19 ; −51 |
| *Bleu mineral* [Mineral blue] | Prussian blue $\mathbf{Fe_4[Fe}CN_6]_3.14H_2O$ | | 22 ; −2 ; −23 |
| *Bleu de Prusse* [Prussian blue] | Prussian blue $\mathbf{Fe_4[Fe}CN_6]_3.14H_2O$ | | 5 ; 4 ; −15 |
| *Sans nom bleu* [No name blue] | Prussian blue $\mathbf{Fe_4[Fe}CN_6]_3.14H_2O$ | Gypsum $\mathbf{Ca}SO_4.2H_2O$ or alunite $\mathbf{KAl}_3(SO_4)_2(OH)_6$ | 17 ; 4 ; −17 |
| *Bleu de Ciel* [Sky blue] | Prussian blue $\mathbf{Fe_4[Fe}CN_6]_3.14H_2O$ | | 34 ; −4 ; −23 |
| *Laque Bleue de Garance Inaltérable* [Blue madder lake] | Prussian blue $\mathbf{Fe_4[Fe}CN_6]_3.14H_2O$ | | 35 ; −5 ; −23 |
| *Cobalt* | $\mathbf{Al_2Co}O$ or $\mathbf{Co}CO_3$? | Hydrocerussite $\mathbf{Pb}_3(CO_3)_2(OH)_2$ or leadhilite $\mathbf{Pb}_4\mathbf{S}O_4(CO_3)_2(OH)_2$ | 40 ; 8 ; −30 |

### 3.1.4. Green Powders

Among the raw green powders (Table 4), we find mainly copper arsenates such as *Vert de Scheele* (Scheele's green), but also an oxide of chromium (eskolaite) (SI1). It was in 1778 that the chemist Carl Wihelm Scheele invented the green pigment copper arsenite, or Scheele's green. In 1814, a more concentrated but very toxic variant was developed, Schweinfurt green [24]. One can note the absence of verdigris ($Cu(CH_3COO)_2 \cdot [Cu(OH)_2]_3 \cdot 2H_2O$) or *Vert de chrome* (Chrome green, a mixture of chrome yellow $PbCrO_4 \cdot PbSO_4$ and of Prussian blue) in the powders used by Fleury Richard, although they had been used for a long time in painting.

In terms of detection, the Scheele's green pigment does not show any peaks in XRD, while the second pigment of the same composition has a signal in XRD. Thus, Raman spectroscopy was shown to be more sensitive, even if a too-long exposure under beam burns the compound.

**Table 4.** Raw green powders (pigments and additives) and their identification (in *italic* font: names written on the parcel papers; in normal font: names given by the authors in the absence of mentions; in **bold** font: the elements detectable by XRF; see SI1).

| Name | Pigment | Additive | L*; a*; b* |
|---|---|---|---|
| Sans nom vert [No name Green] | Arsenolite $\mathbf{As}_2O_3$, trippkeite $\mathbf{CuAs}_2O_4$, emerald green $C_2H_3\mathbf{As}_3\mathbf{Cu}_2O_8$ | | 60 ; −49 ; 20 |
| *Vert de Scheele* [*Scheele's green*] | $\mathbf{Cu}(\mathbf{As}O_2)_2$ or tyrolite $CaCu_5(\mathbf{As}O_4)_2(CO_3)(OH)_4.6H_2O$ | ? | 63 ; −22 ; 21 |
| Sans nom vert foncé [No name dark green] | Eskolaite $\mathbf{Cr}_2O_3$ | Hydrocerussite $2\mathbf{Pb}CO_3 \ \mathbf{Pb}(OH)_2$ | 18 ; −7 ; 13 |

### 3.1.5. Brown Powders

The brown powders (Table 5) are mixtures based on hematite and additives of silica and calcium phosphate, whitlockite for *Terre de Cologne* (Cologne earth) (Figure 2d), plus montmorillonite for *Terre de Sienne* (Sienna) (SI1). The presence of carbon is detected by Raman (SI1) for *Terre de Cologne* (Cologne earth), which is darker than the *Terre de Sienne* (Sienna). The Momie powder appears to be a bitumen (Figure 2c), classically used at the time of the painter's practice. As we can read in a Roret manual [25] dedicated to color manufacturers, "Since we have succeeded in obtaining bitumen in a state of purity, in making it dry and easy to grind, we make much less use of mummy and Van Dyck's brown, bituminous preparations altered by other coloring matters, and which were used almost exclusively in the past". No mineral crystalline phase is detected for the *Couleur d'Appiani* (Appiani's color).

**Table 5.** Raw brown powders (pigments and additives) and their identification (in *italic* font: names written on the parcel papers; in **bold** font: the elements detectable by XRF; see SI1).

| Name | Pigment | Additive | L*; a*; b* |
|---|---|---|---|
| *Terre de Cologne (Mr. Dechazelle)* [*Mr. Dechazelle's Cologne earth*] | Carbon C, hematite $\mathbf{Fe}_2O_3$ | $\mathbf{Si}O_2$, whitlockite $\mathbf{Ca}_3(PO_4)_2$ | 18; 21; 19 |
| *Terre de Sienne brulée Antheaume* [*Burnt Sienna*] | Hematite $\mathbf{Fe}_2O_3$ | Clay montmorillonite | 25 ; 37 ; 32 |

**Table 5.** *Cont.*

| Name | Pigment | Additive | L*; a*; b* |
|---|---|---|---|
| *Couleur d'Appiani* [*Appiani's color*] | Clay? | | 35 ; 32 ; 36 |
| *Momie* [*Mummy*] | Canadian balm (bitumen?) | Silica $SiO_2$ | 25 ; 19 ; 22 |

### 3.1.6. Black Powders

Black powders (Table 6) are mainly carbonaceous bases: burnt humic earth (*Terre de Cassel*/Cassel earth), vine charcoal (*Noir de Vigne*), vine black or bone (or ivory) (*Noir d'Ivoire* and *Grand Noir*/Great black) [26], all of which are obtained by combustion in reducing media. Bone or ivory blacks are characterized by the presence of whitlockite, $Ca_3(PO_4)_2$ (Figure 2d) produced by high-temperature (800 °C) dehydroxylation of the bone apatite $Ca_5(PO_4)_3(OH)$ [27,28]. Through FTIR, the Cassel earth revealed the unexpected presence of walnut brown (melanin-like pigment), which is extracted from walnut shell [29]. We can note the absence of manganese oxide ($MnO_2$) and *Noir de Mars* (Mars black, based on FeO) in the powders used by Fleury Richard.

**Table 6.** Raw black powders (pigments and additives) and their identification (in *italic* font: the names written on the parcel papers; in **bold** font: the elements detectable by XRF; see SI1).

| Name | Pigment | Additive | Picture |
|---|---|---|---|
| *Noir de vigne* [*Vine charcoal*] | Carbon C | Calcite $CaCO_3$, quartz $SiO_2$ | |
| *Noir d'ivoire antheaume* [*Ivory black*] | Carbon C | Whitlockite $Ca_3(PO_4)_2$ | |
| *Grand noir (de Mr)* [*Great black*] | Carbon C | Whitlockite $Ca_{9.5}MgP_7O_{28}$ | |
| *Terre de Cassel* [*Cassel earth*] | Walnut brown (melanin-like) | Silica $SiO_2$ | |

### 3.1.7. White Powders

White powders (Table 7) are mainly composed of lead carbonate and do not have additives. Lead whites are pigments that were synthesized since antiquity and were used for their high covering power until the 19th century, despite their toxicity [30]. Depending on the method of manufacture, the ratio between the amount of cerussite and hydrocerussite can vary; in particular, precipitated *Blanc de plomb* (lead white) pigments are richer in neutral carbonate than those obtained by direct oxidation of the metal [31]. Calcite is also a natural pigment from chalk used since prehistoric times and is one of the first artificial pigments. Its running power is less than lead white [32]. The last pigment, titled "white", appeared to be ground "wine tartar", although we could not find any reference to its use in paint. This choice could be a personal initiative of the artist to use other materials or a counter to a merchant. We can note the absence of *Blanc de zinc* (Zinc white, ZnO), a synthetic pigment present at the time of Fleury Richard but little used in the artistic practice of painting. A work by Johann Georg von Dillis, *Triva Castle* (Bayerische Staatsgemäldesammlungen, Munich), from 1797 reveals its use, for example [33].

**Table 7.** Raw white powders (pigments and additives) and their identification (in *italic* font: names written on the parcel papers; in normal font: names given by the authors in the absence of mentions; in **bold** font: the elements detectable by XRF; see SI1).

| Name | Pigment | Additive | Picture |
|---|---|---|---|
| *Blanc léger* [Light white] | Cerusite **Pb**CO$_3$, hydrocerussite 2**Pb**CO$_3$, **Pb**(OH)$_2$ or plumbonacrite **Pb**$_5$(CO$_3$)$_3$O(OH)$_2$ | | |
| *Blanc de Cremnitz* [Cremnitz white] | Hydrocerussite 2**Pb**CO$_3$, **Pb**(OH)$_2$ | | |
| *Blanc d'argent (de Bellot)* [Silver-white] | Hydrocerussite 2**Pb**CO$_3$, **Pb**(OH)$_2$ | | |
| *Blanc* [White] | Potassium hydrogenotartrate "wine tart" C$_4$H$_5$KO$_6$ | | |
| *Grosse pierre blanche* [Big white pebble] | Calcite **Ca**CO$_3$ (chalk) | Quartz **Si**O$_2$ | |

### 3.2. The Paintings

Non-invasive XRF analysis of paintings simultaneously detects the constituents of all of the painting layers, including: the canvas, the layers of oil-based primer paint, the pigment layer and associated additives, the liquid organic binder (linseed oil, casein, wax, egg . . . ), and finally, the varnish (natural resin, copal, mastic, dammar). The characterized paintings are presented in Figures 3–5 with the analyzed zones highlighted by circles.

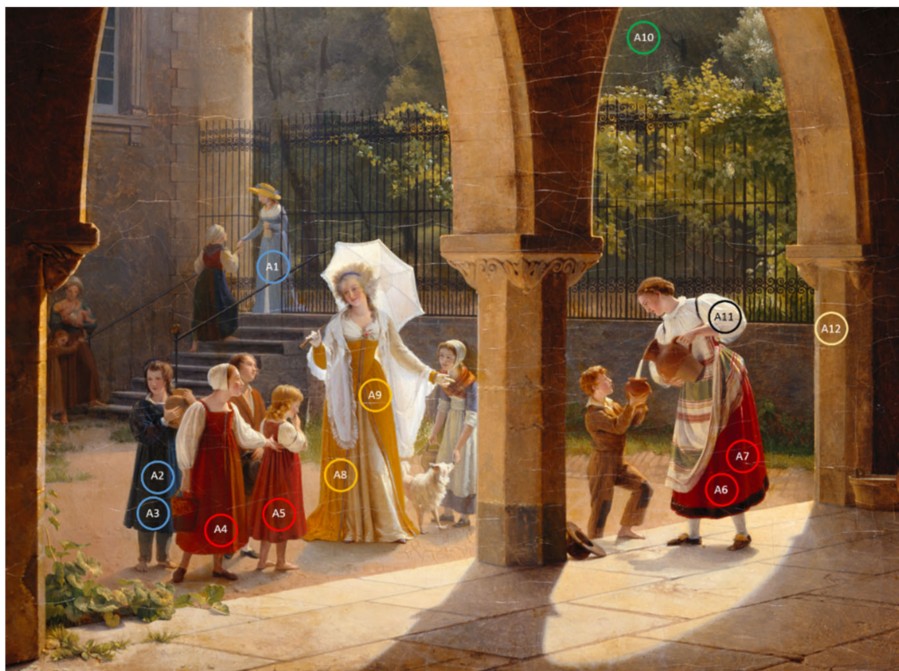

**Figure 3.** Painting *Princess Elizabeth, sister of King Louis XVI, distributing milk* (Fleury Richard, c. 1816), with positioning of the XRF analysis zones (© Musée des Beaux-Arts de Lyon).

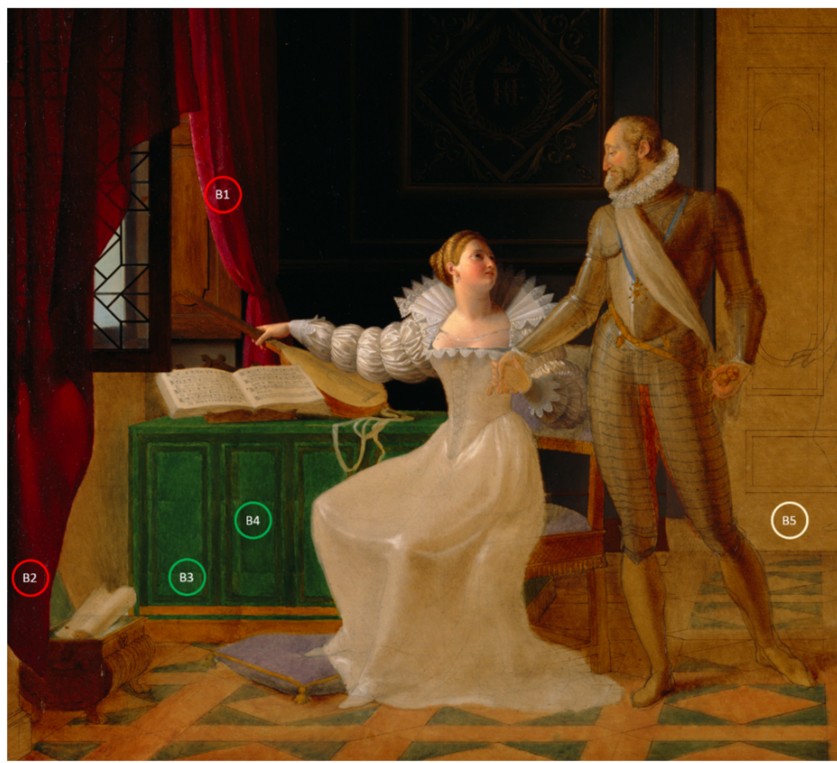

**Figure 4.** Painting *Henri IV at Gabrielle d'Estrées* (Fleury Richard, c. 1810–1812), with positioning of the XRF analysis zones (© Musée des Beaux-Arts de Lyon).

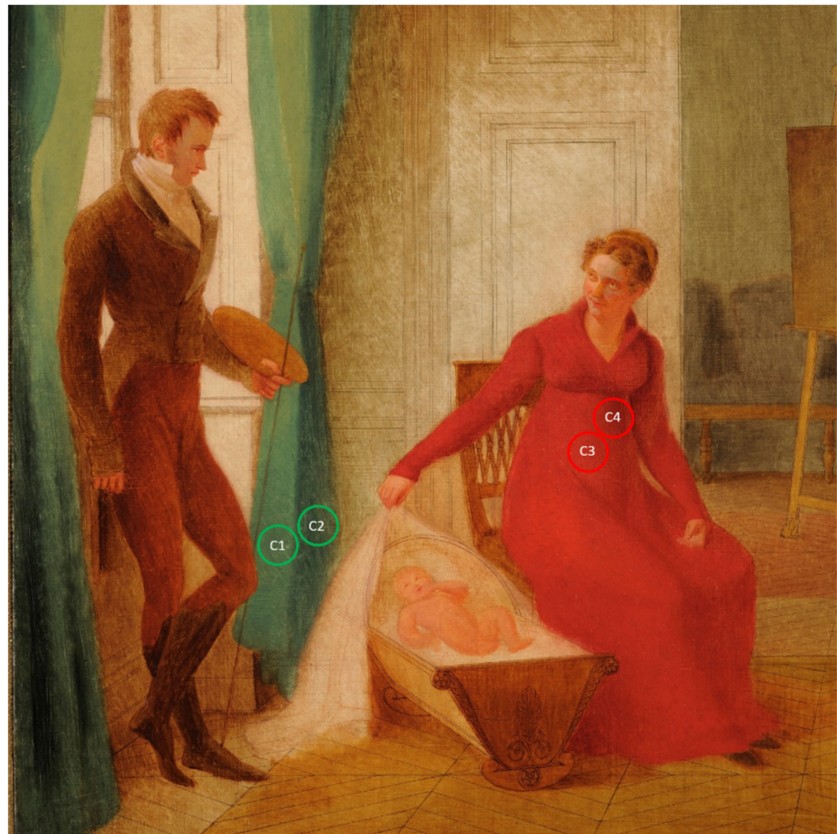

**Figure 5.** Painting *The Painter and his family* (Fleury Richard, 1817 or 1822), with positioning of the XRF analysis zones (© Musée des Beaux-Arts de Lyon).

### 3.2.1. Layers of Oil-Based Primer Paint

The white or lightly pigmented areas (Figure 6a) all show the presence of lead Pb alone, characteristic of ceruse (cerussite $PbCO_3$ or hydrocerussite $Pb_3(CO_3)_2(OH)_2$ used as a primer covering the canvas support of the painting (zone B5, Figure 4). This layer of oil-based primer paint will therefore be systematically present and detected on the entire painted canvas. Ceruse is also the pigment of the areas painted in white (for example, for the sleeve of the milkmaid; area A11, Figure 3).

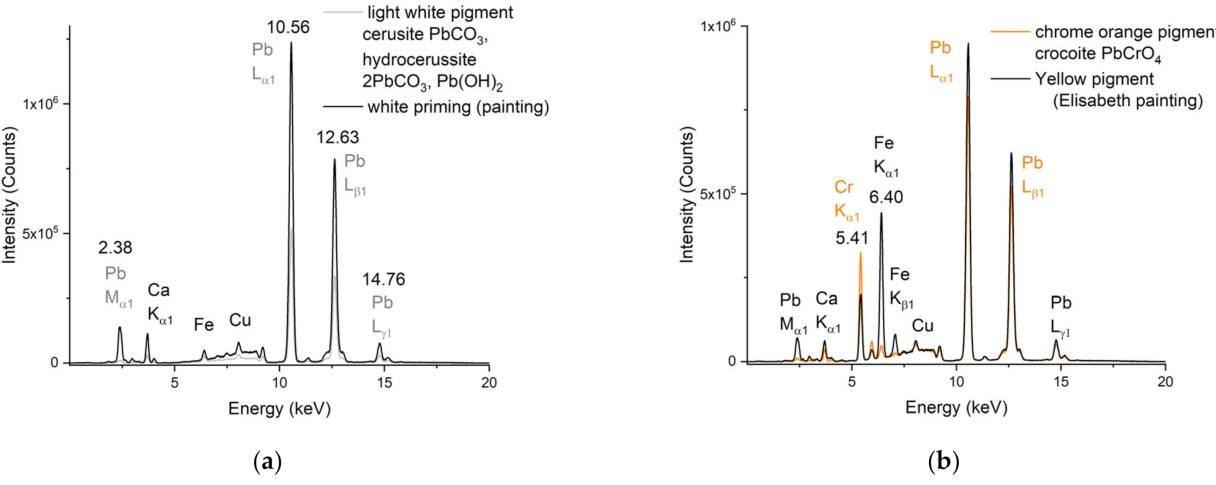

**Figure 6.** Comparative XRF spectra of (**a**) the white primer layer (wall, B5, Figure 4) and the raw Light white pigment; (**b**) the yellow painting layer (Elizabeth's dress, A8 and 9, Figure 3) and the Chrome orange pigment.

We can also note the presence of the element iron (Fe) on all the XRF spectra of the different areas measured on the tables. This reveals the very high sensitivity in XRF to Fe, even in traces.

### 3.2.2. The Yellow-Orange Areas

The presence of chromium makes it possible to highlight the crocoite ($PbCrO_4$) and the Cr-bearing pigments (Vienna Yellow, Lemon chrome, or Chrome orange) and to differentiate them from the primer (Pb alone). The detection of iron in the same yellow areas shows the probable use of a mixture of Yellow or Ru ochre (goethite FeO(OH)) with the crocoite-based pigments ($PbCrO_4$) (Figure 6b; Table 1). Arsenic, which is characteristic of the orpiment and the no name "brown" yellow powders (orpiment $As_2S_3$), was not detected on the yellow areas of the painting *Princess Elizabeth* (Figure 3).

### 3.2.3. The Red Zones

All the red areas analyzed are characterized by the presence of mercury (Hg) and sulfur (S) (Figure 7a), which correspond to vermillion or cinnabar HgS (*Vermillon* or *Vermillion from China*) pigments. The simultaneous detection of iron (Fe) (Figure 7b) reveals the potential use of hematite (pigments Antheaume's English red brown or Mars Red) in mixture with vermilion. An example of these results is given Figure 7a, where the XRF spectrum of zone A6 or A7 of the red skirt (*Princess Elizabeth*) is compared to that of the Vermillion raw red pigment. It should be noted that the relative intensity of the Fe and Hg XRF peaks is variable depending on the areas analyzed. Although the presence of organic pigments (Madder or Beautiful carminic lakes) could not be demonstrated by XRF on the red areas of the various paintings, it is likely that the painter used them in mixture with the other red pigments at his disposal to create a variety of shades in this range of hues (Figure 8).

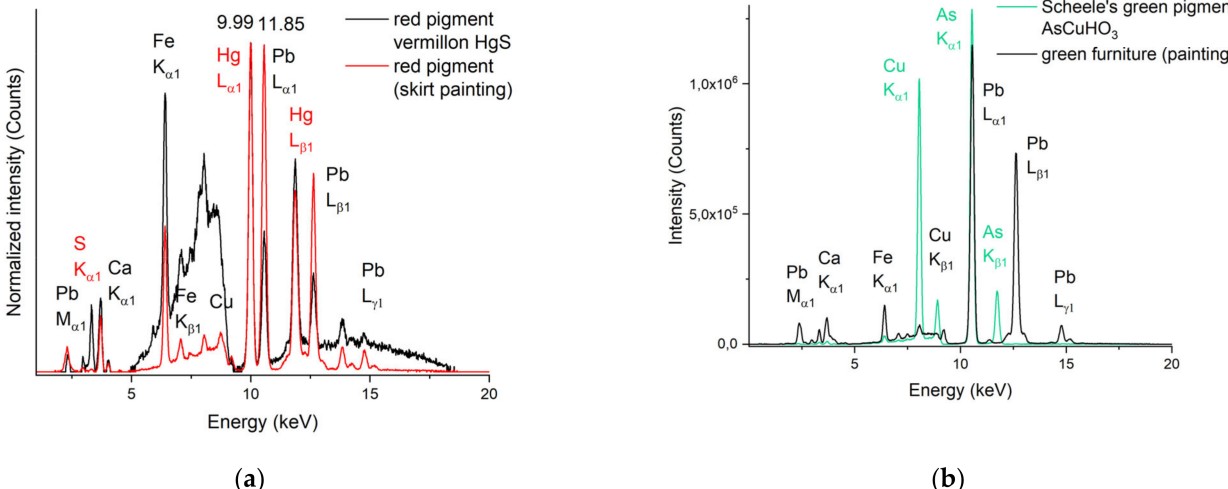

(a)    (b)

**Figure 7.** Comparative XRF spectra of (**a**) the red painting layer (skirt of the milkmaid, A6 and A7 zones, Figure 3) and the Vermillion raw red pigment; (**b**) the green paint (green furniture, B2 and B4, Figure 4) and the raw green Scheele's Green pigment.

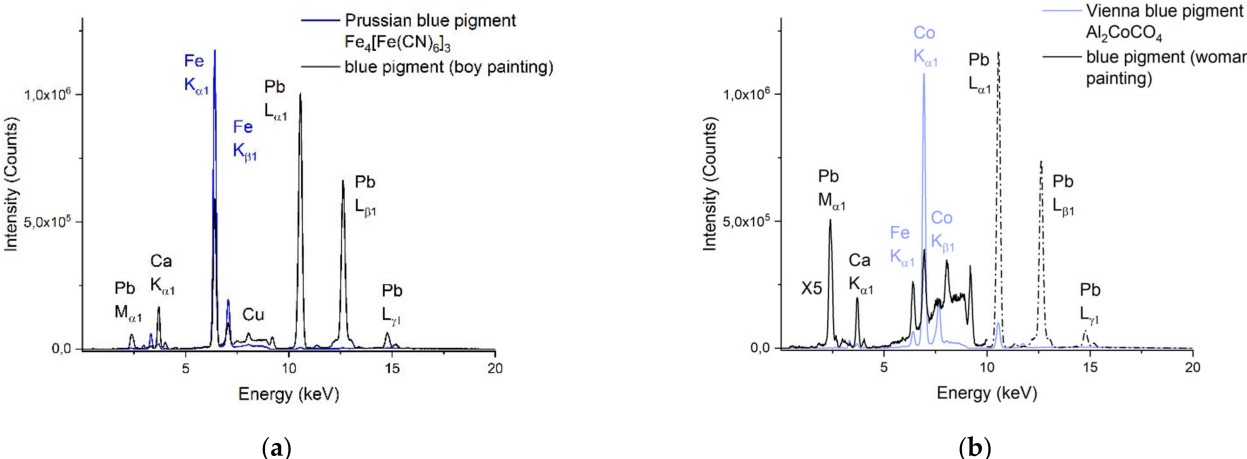

(a)    (b)

**Figure 8.** Comparative XRF spectra of (**a**) the blue painting layer (boy, A2 and A3, Figure 3) and raw Prussian blue pigment; (**b**) the painting layer (blue lady, A1, Figure 3) and the raw Vienna blue pigment.

### 3.2.4. The Blue Zones

The presence of iron (Figure 8a) allows the identification of Prussian Blue ($Fe_4[Fe(CN)_6]_3 \cdot 14H_2O$) in Blue pigments (Prussian blue, Mineral blue, Sky blue or Unalterable blue madder lake) (Table 3) on the blue areas A2 and A3 of Figure 3. The presence of cobalt (Co) being detected on the lilac dress of the woman in the background of *Princess Elizabeth* (Figures 3 and 8b) allows the identification of Vienna blue ($Al_2CoCO_4$).

### 3.2.5. The Green Zones

Arsenic (As) was detected on the paintings by X-ray fluorescence. Its analysis on the paintings is not easy because its $K_{\alpha 1}$ line (the most intense one) is superimposed with the $L_{\alpha 1}$ line of lead. The fixed ratio between the $L_{\alpha 1}$ and $L_{\beta 1}$ lines of lead allows for highlighting the possible contribution of the $K_{\alpha 1}$ of arsenic on the first line. The green areas of the tables are thus characterized by the simultaneous presence of copper Cu and As (Figure 3), which reveals the use of Scheele's green ($AsCuHO_3$) or Emerald green (Figure 7b). This hypothesis is reinforced by the absence of chromium Cr, which indicates that the eskolaite pigment $Cr_2O_3$ was not used in the analyzed areas of the tables.

### 3.3. Colorimetric Analysis of Raw Pigmented Powders and Paints

The study of colors on paintings is interesting because it is perfectly non-destructive and relatively little time-consuming. It has allowed for completing compositional studies of painting [34,35], understanding the practices in the use of colorants [36], and guiding restoration works [37]. We propose here a comparison of the colorimetric indices of the colored areas of the paintings to those of the pigmented powders (SI2).

Figure 9 represents the hues a* and b* for the different pigments as well as those of the areas measured on the three paintings of Fleury Richard (Figures 3–5). The tints measured on the paintings are globally in good agreement with those of the pigments.

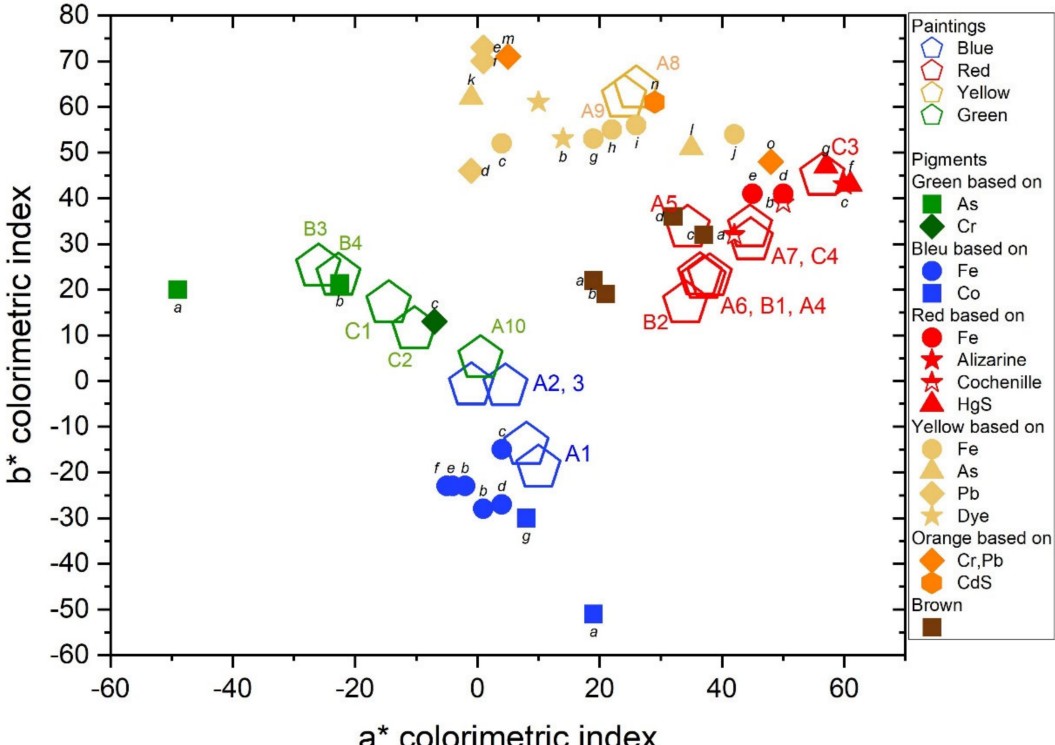

**Figure 9.** Colorimetric index a* and b* measured on the powdered pigments and on the various points of the pictorial layers indicated in Figure 3 (noted A), Figure 4 (noted B), and Figure 5 (noted C). The powders have been grouped by color and by the chemical elements of their coloration; the letters indicate their name as described below in the text. **Green powders**: (a) No name green, (b) Scheele's green, (c) No name dark green; **Blue powders**: (a) Vienna blue, (b) No name blue, (c) Prussian blue, (d) Mineral blue, (e) Sky blue, (f) Madder blue lake, (g) Cobalt; **Red powders**: (a) Madder lake, (b) Red pebble, (c) Magnificent lake, (d) Antheaume's red brown, (e) Mars red, (f) Vermillion, (g) Vermillion from China; **Yellow powders**: (a) Gaude lake, (b) Yellow lake, (c) Light yellow, (d) Small yellow stones, (e) Vienna yellow, (f) Lemon chrome, (g) Yellow stone, (h) Yellow ochre, (i) Ru ochre, (j) Brown yellow, (k) Orpiment, (l) No name brown; **Orange powders**: (m) Chrome orange, (n) Cadmium (orange), (o) No name orange; **Brown powders**: (a) Momie, (b) Cologne earth, (c) Burnt Sienna earth, (d) Appiani's color.

For **yellows and oranges**, the range of shades is rather varied (SI2). The pure yellows correspond to Vienna yellow, Lemon chrome, Yellow Orpiment, Small yellow stone and Light yellow. The two lead-based pigments have the largest tones (b* larger). All other yellow pigments have a red component and therefore appear in the same colorimetric area as the orange pigments. In Figure 9, the two analyzed areas corresponding to the yellow-orange dress of Elizabeth (Figure 3) appear as expected in this zone. As chromium is detected in XRF, the pigment used is most probably orange chromium.

For **reds**, the colorimetric indices of red pigments are placed in the red zone with a yellow tone (b* positive); thus, they are so-called warm colors. The shades are all close, with vermilion being the purest red. The different reds measured on the paintings are in the same range. It is therefore difficult to identify which pigment was used by this technique. Only one point, C3, which corresponds to the dress of the woman in the painting *The Painter and his family* (Figure 5), has an index higher than the ochre. Thus, it is necessarily vermilion based. These results are consistent with XRF analysis (SI1 and SI2).

For **blues**, the Cobalt blues have a red component while the Prussian blues have none or very little (Mineral blue). Points A2 and A3 correspond to the dark dress of the child in the painting *Princess Elizabeth* (Figure 3). Their position on the graph indicates that they are most probably based on Prussian blues. The light blue dress of the lady in the background of the same painting (Figure 3) has a slight red component, which could be due to the presence of the Cobalt blue pigment. These results are in good agreement with those of XRF (SI1 and SI2).

For **greens**, Scheele's greens have a colorimetric index a* greater in absolute value than the chromium-based one (SI1). The colorimetric index b* is positive, which means that the greens include a slight yellow tint, thus a tint commonly known as warm. The colorimetric index of the greens measured on the painting *Henri IV at Gabrielle d'Estrées* (Figure 4), B3 and B4, corresponds to Scheele's green, which is in good agreement with the XRF measurement. The colorimetric index of the greens measured on the painting *The Painter and his family* (Figure 5), C1 and C2, have colorimetric index of absolute values lower than those of Scheele's green but higher than those of chrome green. Since the index of pure pigment is necessarily higher than the index of a pictorial layer made of it, these greens cannot be only Chromium green (unnamed green pigment). Moreover, according to the XRF analysis, there is no chromium detected on the paintings (SI1). Therefore, it is probably Scheele's green mixed with a dark pigment, most likely Fe-based, such as, for example, the Sienna or Cologne earth pigments.

Finally, it should be noted that none of the tints measured on the different paintings are outside the colorimetric zones of the different pigments. This suggests that most of the colored pigments were used pure or with slight mixing. For example, there is no violet in the analyzed paintings.

## 4. Conclusions

The coupling of several structural and spectroscopic analytical techniques proved to be decisive, in most cases, for the accurate determination of pigmented powders contained in Fleury Richard's cabinet. X-ray diffraction and Raman spectrometry allowed for the identification of mixtures of rather mineral and well-crystallized phases with different sensitivity (effective section) depending on the nature of the pigments; X-ray fluorescence proved to be valuable for the identification of pigment trace-elements, while infrared absorption was more sensitive to organic compounds.

Concerning the paintings, only a non-destructive analysis method could be performed—for instance, XRF. The systematic presence of lead carbonates in the priming of the canvases makes it difficult to determine the pigment layers with lead (*Chrome yellow*, *crocoite*), arsenic (*Scheele's green*), or mercury (*Vermillion*), as their X-ray fluorescence lines partially overlap with those of Pb. However, thanks to the detection of other elements and comparison with the analysis of the raw pigments as well as the colorimetric results, we were able to reach a conclusion on whether these pigments were in the paintings or not.

The pigments contained in the color-mixing cabinet are undeniably the same as those used on the canvases, and the analyses confirm that they reflect the palette of Fleury Richard, who left few writings detailing his practice. The use of many pigments that were already ground or even mixed with additives shows that he did not prepare all of his pigments but used the services of his (often Parisian) color merchants. The hypothesis that the added materials could have been included without the buyer's knowledge is consistent with the concerns of the time [38], which show a great distrust of adulterated goods. However,

this remains doubtful in the case of Fleury Richard, not only given the painter's expertise as a colorist and his reputation, but also because of the large number of pigments that are not pure. Indeed, Fleury Richard's work gives a great importance to glazes and light effects likely to emphasize pure colors (colorimetry reveals very few composed colors in the corpus studied) as well as the texture and materiality of the pigments; their fineness and adherence were of undeniable importance. The fact that some pigments were still to be ground shows that the painter did not systematically use prepared pigments and suggests that he chose his products according to specific needs. Similarly, the absence of many pigments that were common in Fleury Richard's time, contrasting with the large number of pigmented powders in the cabinet and the richness of the ochre shades, reveals the painter's singular choices and preferences.

The colorimetric analysis carried out on the canvases reveals that the painter almost never mixed the colors for the realization of the analyzed paintings. This confirms the nature of the chromatic innovations carried out by Fleury Richard, which resided not so much in the elaboration of new colors, which were rather provided to him by his merchant (such as Scheele's green), but in the development of processes that made it possible to emphasize them—for example, through glazing techniques.

The chemical compositions of the pigmented powders invite many hypotheses concerning the real nature of certain pigments. While the addition of alum could improve adhesion, the addition of certain cheaper substances to the pigments would have allowed the merchants to make additional profits.

The analysis of all of the results obtained through the different techniques used provides us information not only on the painter's work and the type of raw materials he had access to, but also on the work of color merchants at a time when their profession was experiencing a period of transformation. The study of the contents of Fleury Richard's cabinet thus underlines the complexity and challenges of his work as a colorist.

**Supplementary Materials:** The following supporting information can be downloaded at: https://www.mdpi.com/article/10.3390/heritage5020066/s1. Spectral data of the pigmented powders (SI1) and Colorimetric index measurements (SI2).

**Author Contributions:** Conceptualization, E.W.; historical analysis, E.W. and S.P.; formal analysis, D.C., A.B.-L., C.L.L., A.P. and G.P.; investigation, A.B.-L.; methodology, E.W. and S.P.; writing—original draft, D.C., S.P., A.P. and G.P. All authors have read and agreed to the published version of the manuscript.

**Funding:** This research received no external funding.

**Institutional Review Board Statement:** Not applicable.

**Informed Consent Statement:** Not applicable.

**Data Availability Statement:** Not applicable.

**Conflicts of Interest:** The authors declare no conflict of interest.

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
