# Peer review of "The Pigments of the Painter Fleury Richard (1777–1852), a Model for Multidisciplinary Study"

_heritage, doi:10.3390/heritage5020066_

Round 1

Reviewer 1 Report

The objective and the samples analyzed are interesting, which perfectly justifies submitting a manuscript reporting their analysis to the journal Heritage. On the other hand, the manuscript does not correspond to what is expected for the journal. The experimental justifications of the assigned phases as present in the samples are not or are very little proven. Further, the authors ignore the extensive literature using vibration spectroscopy, XRD and X-ray fluorescence to identify all of the ingredients present. Some major publications to cite are listed below but there are many more. Furthermore, the authors are unaware that much of the previous work has shown that the combination of XRD, IR, Raman and X-ray fluorescence is often insufficient to identify all the ingredients and that Mass spectroscopy and chromatography must also be used. Authors should provide in the text or in the Appendix more sets of spectroscopic proofs. The tables must be more precise, specifically listing the major and minor elements identified for each color, the phases identified in XRD, Raman, etc. and the proposed attribution (knowing that not all phases can be identified by the simple Raman and XRF conjunction, IR analysis being generally essential for organic pigments).

Information is missing in the experimental description: surface/quantity analyzed in XRF, XRD, FTIR/Raman, colorimetry

Ricci et al., J Raman Spectrosc. 2004, 35(8-9), 616 ; Rosi et al., J Raman Spectrosc. 2004, 35(8-9), 610 ; Schulte et al., J Raman Spectrosc. 2008, 39(10), 1455 ; Ropret et al., J Raman Spectrosc.2008, 69(2), 486 : Scherrer et al. Spectrochim Acta Part A 2009, 73(3), 505 ; Stanzani et al. Virb. Spectosc. 2016, 85, 62 ; Mulholland et al. Heritage Sci 2017, 5, 43 ; Etc.

Author Response

Please find attached the answer to your comments

Reviewer 2 Report

This paper combines two great aspects in studying the work of a particular artist, that are the characterization of its materials and the access to study the artist’s artifacts at the same time. In the framework of this, I believe that this paper does belong to the theme and scope of Heritage, but after taking into consideration some things.

The strong points of this work are that it is dealing with the palette of a famous painter, whose work represents the chronological transition -or better the incorporation- of chemical manufactured pigments, and the authors compare it with three of the painter’s works. The authors use well-established and complementary techniques for their measurements. Regarding their findings, the absence of lead tin or Naples yellows in the artist’s palette is a very interesting thing, that can may be used for authentication reasons. This applies to the absence of some other pigments as the authors have noticed. I should also mention that the authors make a very good use of language, and the presentation of their results is very organized and easy to follow.

My strong objection to the present manuscript has to do with the absence of presentation of certain measurements. The authors use XRD, FTIR, XRF, Raman and UV-Vis spectrophotometry for the identification of the pigments, but -in the manuscript- everything has to do only with xrf and uv-vis. Not a single spectrum or diffractogram is present or even commented on. Following this, problems regarding the attribution between two different pigments could be easily overcome with Raman, FTIR or even XRD. This is very confusing, as impression is derived that these techniques were not applied at all.

In the following, I have some points that should be taken into consideration:

  1. Abstract and/or keywords: Please, do mention the analytical techniques used for the identification of the pigments
  2. 143. Please, correct to “Fourier transform infrared spectroscopy”. The same applies to L. 172.
  3. 171. Acquisition time 192 seconds isn’t a little bit short?
  4. 172-181. Apart from the previous comment, the authors state that they used a blank reference KBr pellet, in transmittance mode. So, assuming that the KBr pellet technique was applied,
  5. 200-212. I find very correct the fact that the authors compare the pigments and the paintings with the same measurement method, as UV-Vis spectrophotometry is a very subjective method in the case of comparing 2 different measurements that were conducted with different instruments or even methods. For this reason, did the authors use the same illumination conditions during the photography of both pigments and paintings? Is the presence of varnish on the paintings taken into consideration?
  6. Line 231 and all tables. Although it is clear to understand what the authors want to say, “medium” is usually used meaning “binder”. Inert pigment, extender, filler or additive are commonly used to describe this situation. The authors can refer to “R.J. Gettens, G.L. Stout, Painting Materials. A short encyclopaedia, Dover Publications, Inc., New York, 1966”, and “N. Eastaugh, V. Walsh, T. Chaplin, R. Siddall, The Pigment Compendium A dictionary of historical pigments, Elsevier Butterworth-Heinemann, Oxford, 2004” for proper definition.
  7. 241-243. XRD, Raman and FTIR spectra and patterns should be presented and the peaks attributed to. Regarding jaune clair, it is quite strange that no identification was done. The measurements’ presentation could -maybe- be useful to potential readers.
  8. Table 3. No bold fonts are present, indicating the XRF detection. The same for Table 5
  9. 245-256: XRD patterns and FTIR/Raman spectra should be provided and attributed.
  10. 277-288: XRD patterns and FTIR/Raman spectra should be provided and attributed. Additionally, the identification of Prussian blue cannot be characterized as “a delicate mater” when FTIR is one of the applied techniques, this is an exaggeration. The ~2088 cm-1 peak is so strong that it is clearly present even in cases where elemental Fe is almost at detection limit through EDS or XRF.
  11. 301-304. Scheele’s and emerald green can both be detected from FTIR spectra. The collected spectra and patterns are missing.
  12. 309-318 and Table 5. It’s very interesting that no umbers are detected (absence of MnO2) Spectra? Patterns?
  13. Table 6. What is melanin?
  14. XRD, FTIR and Raman measurements are also absent from the rest of the pigments.
  15. 352-355. This sentence should be rephrased as it can be misleading. Firstly, canvas and varnish results are not presented. Moreover, what’s the penetration depth of XRF, that all these layers can be identified with non-invasive and surface -I presume- analysis? Finally, by nominating organic binders, varnish and to some point canvas can mislead, as XRF just makes elemental detection.

Some minor language points.

  1. 230. An “et” has slipped here
  2. 328. “We cab note”
  3. 309 & 335. “media”, although another word for these additives should be used.
  4. 352. Detects or detected.
  5. 385. Were not detected
  6. 414-415. Maybe tables instead of tables? Or I am missing something

Author Response

Please find attached the answers to your comments

Reviewer 3 Report

Manuscript ID: heritage-1711378

Title: The pigments of the painter Fleury Richard (1777-1852), a model for multidisciplinary study

  1. Recommendation:

Reconsider after major revisions

  1. Comments to the authors:

2.1 Overview and general recommendation

The paper "The pigments of the painter Fleury Richard (1777-1852), a model for multidisciplinary study" presents an interesting case of study of the artist's Fleury Richard materials, available for study and the comparison of the analytical results obtained from those materials with some of his paintings. The work is certainly of interest to the scientific community, particularly because it reports on the analytical information of original raw materials used by an artist and that can assist in the understanding of analytical results obtained from artwork made at the same time as the artist's activity.

However, the work requires further improvements, particularly, regarding technical aspects and a precise description of the results.

I am confident that if the authors are willing to rearrange and complement the information presented, the work will improve considerably. For this reason, I recommend reconsidering the paper after major revisions. I explain my concerns in detail below.   

2.2 Major comments

One of my major concerns regards the description of the techniques used in the investigation and the way the results are presented. Some technical details are missing and the authors could eliminate some of the generic parts regarding the analytical methods and describe more precisely the analytical set-up and experimental conditions.

Also, there seems to be confusion between the terms dye, pigment, and lake, please revise this terminology throughout the text. Something similar happens with the terms medium and binder.

The colourimetric measurements are also a major concern since the experimental conditions are not precisely reported and it seems different systems were used for the reference materials and the paintings. Moreover, the identification of pigments using only colourimetric values is not the best experimental choice due to all the possible interferences.

2.3 Minor comments

Page 2. Line 56. It seems that the term "medium" used by the authors refers to the fillers while siccative should be substituted by binder.

Page 3. Line 107. Please add a citation for the information regarding the use of pig bladders and Pb tubes.

Page 3. Line 123. Section Materials and Methods. Please consider adding a table with the description of the pigments analysed. In the table, you can add a code for each pigment (e.g., Y1, Y2, R1, etc.) and add there the name written on the paper and the English translation as well as the visible colour image. This could help you to save space in the tables for the results.

Page 4. Lines 128-133. There are three figures mentioned in the text (2-4) but the images are not in the text. The figures should appear after the first mention in the text.

Page 4. Lines 135-148. Please consider eliminating the introductory section to the methods.

Page 4. Section 2.2.1. X-ray fluorescence (XRF). The information reported in this section should be focused on the technical aspects and experimental setup. Did you perform the analysis of the paper as a blank? Some trace elements, such as Fe, Ba, and Ca, could be present as contaminations deriving from the paper production or due to filler and other additives used to improve the paper properties. You could report the results in the supplementary information.

Page 4. Line 170. Please substitute the word theta for the Greek letter.

Page 4. Line 172. Section 2.2.3. Infrared absorption. Please consider substituting this with "Fourier transform infrared spectroscopy (FTIR)."

Page 5. Line 174. Please specify the ratio or concentration of pigment/KBr that you used for the experiments. In the same section please add the measurement parameters: spectral resolution, number of scans, number of scans for the background, and software used for the data processing.

Page 5. Lines 179-181. Please consider eliminating this section.

 Page 5. Line 182. Section 2.2.4 Raman. Please substituted with micro-Raman spectroscopy. In the same section please specify the measurement parameters: power laser in the sample surface, spot size, number of scans, spectral resolution, and other technical information regarding the two instruments you used. It seems both instruments you used are coupled with microscopes, please add the information about the objective you used (i.e., NA, magnification). Please be sure the model of the instruments is correct; you should probably need to add LabRAM to Aramis.

Page 5. Lines 192-212. Section 2.2.5. Colorimetric analysis. These measurements are a major concern regarding the methodology used in this work. More technical details are required, please report the precise experimental set-up you used. For example, please include the type of illumination you used and possibly the geometry employed between the illuminant and camera. You mentioned. Please indicate the instrument used for the colourimetric measurements of powdered pigments (model, brand, etc.), you also indicate the use of paper as white. Why did you decide to use that? In general, a white standard (spectralon) is used to measure the white. Did you also measure the black? Please indicate.

You mention you used a pigment layer but without a binder, since you compared your results with a real painting it would be better to measure the values of a paint mock-up considering also the binder that certainly influences the final hue of the paint layer and it accounts for the surface topography, this is the closest system to your case of study.

Please add the pixel size of your images and the number of pixels you consider for each area measured, please also indicate any possible correction you used (e.g., segmentation).

Page 5. Line 211. Please indicate why you did not measure the white and black areas.

Page 5. Lines 214-216. Section Results. Consider eliminating the introductory phrases.

Page 5. Line 217. Section 3.1. Raw pigments. Please consider substituting "Raw" with "Powder"

Page 6. Table 1. The table appears before the first mention. Also, for Table 1 and all the tables containing data results. The way the information is presented is not clear. I assume the information reported was obtained by putting together all the results from the different techniques used. You should specify the information obtained with every technique in separated columns, for example, one column for XRF data, one for FTIR, one for Raman, and one for colourimetric coordinates. If you add a table in the methods section reporting the name of samples and photos, you can specify the analytical results in the tables presented in the Results section.

For the colourimetric values, please report also the standard deviation of each value calculated.

Page 6. Line 228. Please consider substituting the word "crude" with "raw" or "powder" pigments. This should be done through all the paper.

Page 6. Line 230. There is “et” instead of “and”. Please change it.

Page 6. Line 231. It seems that the word “filler” could be a better option for “medium.”

Page 7. Lines 249-256. Section Red pigments. Please be more precise with the information regarding lake pigments. In line 249 you refer to the madder dye as a "pigment" which is incorrect terminology. The dye is extracted from the roots and later the lake is obtained by precipitating the dye in an inorganic substrate (alum, calcite, etc.). Please also notice that Alizarin is one of the main components of madder lake, there are other abundant markers such as purpurin. Does Raman or FTIR analysis give clues about the inorganic substrate? You should be able to identify them with one of those techniques.

Page 8. Line 268. What does “(Cobalt)” refers to? It does not seem to be linked to the phrase. Please specify in the text.

Page 8. Line 280. When discussing Raman or FTIR results, please specify the wavenumbers of the characteristic bands and their assignments.

Page 8. Line 284. Have you considered the blue pigment Smalt? It contains Co but does not exhibit crystalline phases in XRD. However; you should be able to identify some silicates in FTIR or Raman. In the table you report silica, so please revise this interpretation.

Page 9. Lines 296-304. According to the results reported in your table (which I assume you obtained using Raman), you report the presence of arsenolite, please discuss a little bit on this, since it indicates that your green pigment, which is already degraded, must probably by photooxidation.

Page 10. Line 326. Did you identify the characteristic signals of the phosphates? Please be more precise. Also, in line 328 it should say “can” instead of “cab.”

Section 3.2. Paintings. In the XRF results from the paintings, since Pb and Fe are present in many areas and probably are related to the presence of those pigments in different layers of the painting stratigraphy, please consider calculating counts rations with other elements present in all the points analysed, for example, Ca show in a more precise way the points in white the number of counts is higher and thus suggest the presence of a pigment containing those elements, this can help in the white areas regarding the presence of lead white, and in the red and blue areas regarding the presence of Fe oxide and Prussian blue pigments.

3.3. Colorimetric analysis of raw pigments and paints. This section is particularly problematic since it seems you used two different systems to measure the colourimetric values and the measurements do not account for the influence of the binder. In addition, the results do not agree, for example, in the scatter plot in figure 8, only two of the yellow areas from the painting are reported and are closer to the distribution of CdS and Ru ochre but XRF results indicate a different composition, also, in the discussion there are more than two areas mentioned.

Page 14. Section 3.2.4. The blue zones. You suggest the use of Vienna blue due to the presence of Co. Why did you discard the possibility of Smalt?

Page 16, lines 478-481. You mentioned that the pigments were used pure and not in a mixture; however, in the XRF section, you suggest some mixtures, for example, cinnabar, Fe oxide, and probably lakes pigments in red areas. You also suggest the presence of Lead white in lightly pigmented areas, please revise this.

Author Response

(The authors gave the same response as above.)

Round 2

Reviewer 1 Report

The authors have clarified/completed the text as requested

Author Response

The authors have clarified/completed the text as requested.

Thanks a lot for your time and contribution to improve our manuscript.

Reviewer 2 Report

Dear authors, thank you for your response, I have some points.

11.       Regarding my remarks about the absence of Raman, FTIR and XRD measurements. It’s common sense that NOT all measurements can or should be presented in a single paper. In all cases regarding this kind of studies, such measurements are shown by representatives, as the complete absence of measurements can be a little suspicious. I hope I made myself clear. I still insist on the presentation of some representative FTIR, Raman and XRD measurements inside the main body of the manuscript, as this kind of presentation makes the work to seem that it is mainly based on colorimetric facts.

22.       Abstract. Thank you for incorporating the applied analytical techniques. Please, avoid to not explain abbreviations upon their first mentioning, this applies not only for the main body of the manuscript, but for the abstract also.

33.       Acquisition time of XRD measurements. Please, check again your measurement parameters, as something is off. Step size of 0.0205° for a range of 65° and ~3 minutes measurement leads to 0.06 s per step which is very very short, even for a continuous measurement. “the large number of pigments” is not an answer.

44.       Line 200. Spectroscope, not spectroscopy, as it refers to the instrument.

55.       Line 204. Technique

66.       Melanin as a term refers to biological skin pigment. The pigment is usually referred as “hypocastanum or chestnut brown”. If the authors insist on this term, this should be explained and referenced, as it can be misleading.

Author Response

Dear authors, thank you for your response, I have some points.

  1. Regarding my remarks about the absence of Raman, FTIR and XRD measurements. It’s common sense that NOT all measurements can or should be presented in a single paper. In all cases regarding this kind of studies, such measurements are shown by representatives, as the complete absence of measurements can be a little suspicious. I hope I made myself clear. I still insist on the presentation of some representative FTIR, Raman and XRD measurements inside the main body of the manuscript, as this kind of presentation makes the work to seem that it is mainly based on colorimetric facts.

We add a figure 2 with 4 representative spectra obtained by XRD, FTIR and Raman

  1. Abstract. Thank you for incorporating the applied analytical techniques. Please, avoid to not explain abbreviations upon their first mentioning, this applies not only for the main body of the manuscript, but for the abstract also. Done
  2. Acquisition time of XRD measurements. Please, check again your measurement parameters, as something is off. Step size of 0.0205° for a range of 65° and ~3 minutes measurement leads to 0.06 s per step which is very very short, even for a continuous measurement. “the large number of pigments” is not an answer.

We are conscious that the XRD durations could have been much higher. However only a few quantity of raw pigments were available to be characterized by XRD. Some with longer durations in case of failure of the identification didn’t bring additional information. Furthermore, it was not possible to retrieve the tiny quantity to run them a second time with longer exposure. This was balanced by the use of the other analytical techniques.

  1. Line 200. Spectroscope, not spectroscopy, as it refers to the instrument. Done
  2. Line 204. Technique Done
  3. Melanin as a term refers to biological skin pigment. The pigment is usually referred as “hypocastanum or chestnut brown”. If the authors insist on this term, this should be explained and referenced, as it can be misleading.

The Cassel earth reveals by FTIR the unexpected presence of walnut brown (melanin) which is extracted from walnut shell ‎[29]

Yao, Z.; Qi J.; Wang, L. Isolation, fractionation and characterization of melanin-like pigments from chestnut (Castanea mollissima) shells. J Food Sci. 77(6), 2012, p. C671-C676.

Thanks a lot for your time and your accurate questions which contributed to improve our manuscript.

Reviewer 3 Report

I want to thank the authors for considering the previous comments. There are still some issues, mainly regarding the English language that require attention. For this reason, I suggest accepting the article after those minor revisions. Here below are some of the issues identified:

 Page 2, line 58. it should say an additive, a binder, and a siccative.

Page 5. Line 199.  you can use only KBr or consider modifying to "potassium bromide (KBr) powder"

Page 5. Line 204 should say "technique" instead of "technic". In the same line, you should use the minus sign (–) instead of the hyphen (-) for cm–1

Author Response

I want to thank the authors for considering the previous comments. There are still some issues, mainly regarding the English language that require attention. For this reason, I suggest accepting the article after those minor revisions. Here below are some of the issues identified:

 Page 2, line 58. it should say an additive, a binder, and a siccative. Done

Page 5. Line 199.  you can use only KBr or consider modifying to " potassium bromide (KBr) powder " Done

Page 5. Line 204 should say "technique" instead of "technic". In the same line, you should use the minus sign (–) instead of the hyphen (-) for cm–1 Done

Thanks a lot for your time and your accurate reading and remarks allowing us to improve our manuscript.